# Discrete Choice Analysis of Consumer Preferences for Meathybrids—Findings from Germany and Belgium

**DOI:** 10.3390/foods10010071

**Published:** 2020-12-31

**Authors:** Adriano Profeta, Marie-Christin Baune, Sergiy Smetana, Keshia Broucke, Geert Van Royen, Jochen Weiss, Volker Heinz, Nino Terjung

**Affiliations:** 1DIL e.V.–German Institute of Food Technology, Prof.-von-Klitzing-Straße 7, D-49610 Quakenbrück, Germany; m.baune@dil-ev.de (M.-C.B.); s.smetana@dil-ev.de (S.S.); v.heinz@dil-ev.de (V.H.); n.terjung@dil-ev.de (N.T.); 2Institute for Agricultural and Fisheries Research (ILVO), Brusselsesteenweg, 370, 9090 Melle, Belgium; keshia.broucke@ilvo.vlaanderen.be (K.B.); geert.vanroyen@ilvo.vlaanderen.be (G.V.R.); 3Department of Food Structure and Functionality, Institute of Food Science and Biotechnology, University of Hohenheim, Garbenstraße 21/25, 70599 Stuttgart, Germany; j.weiss@uni-hohenheim.de

**Keywords:** meat substitute, meathybrid, consumer preference, plant-based proteins

## Abstract

High levels of meat consumption are increasingly being criticised for ethical, environmental and social reasons. Plant-based meat substitutes have been identified as healthy sources of protein that, in comparison to meat, offer a number of social, environmental and health benefits and may play a role in reducing meat consumption. However, there has been a lack of research on the role they can play in the policy agenda and how specific meat substitute attributes can influence consumers to partially replace meat in their diets. This paper is focused on consumers’ preferences for so-called meathybrid or plant-meathybrid products. In meathybrids, only a fraction of the meat product (e.g., 20% to 50%) is replaced with plant-based proteins. Research demonstrates that in many countries, consumers are highly attached to meat and consider it as an essential and integral element of their daily diet. For these consumers that are not interested in vegan or vegetarian alternatives as meat substitutes, meathybrids could be a low-threshold option for a more sustainable food consumption behaviour. In this paper, the results of an online survey with 500 German and 501 Belgian consumers are presented. The results show that more than fifty percent of consumers substitute meat at least occasionally. Thus, about half of the respondents reveal an eligible consumption behaviour with respect to sustainability and healthiness, at least sometimes. The applied discrete choice experiment demonstrated that the analysed meat products are the most preferred by consumers. Nonetheless, the tested meathybrid variants with different shares of plant-based proteins took the second position followed by the vegetarian-based alternatives. Therefore, meathybrids could facilitate the diet transition of meat-eaters in the direction toward a more healthy and sustainable consumption. The analysed consumer segment is more open-minded to the meathybrid concept in comparison to the vegetarian substitutes.

## 1. Introduction

There are more than 7.7 billion people on this planet, with forecasts predicting the population to grow to 9.7 billion by 2050 [1]. Securing a sustainable food supply for humankind is therefore becoming a major challenge. Diets with a high share of animal proteins must be adapted in order to ensure that demand is not outstripping production [2,3]. Furthermore, the consumption of meat and meat products in larger portions is associated with higher risk of cardiovascular, coronary and cerebrovascular diseases, stroke, diabetes type 2 and colorectal cancer [4].

In addition to these health issues, meat production chains have a considerable impact on the environment through the use of land, the application of fertilisers, greenhouse emissions and water consumption, resulting in the loss of biodiversity and enhancing climate change [5,6,7,8]. It causes more emissions per unit of energy compared with plant-based foods because energy is lost at each trophic level. Meat production is the most important source of methane, which has a relatively high global warming potential, but a lower half-life in the environment compared with that of CO_2_ [9]. The carbon footprint of plant-based foods on average is twice as low as the impact of pork [10], while the impact in some other categories can be more than 60 times lower [11]. We also highlight that meat and meat products are associated with severe animal welfare issues, such as pigtail docking, poultry debeaking, calve separation and mistreatment in slaughterhouses [12,13].

Integrating new protein sources into the diet as a solution for the mentioned problems means overcoming barriers such as traditional meat consumption across many cultures [14]. Recent research put forward the idea that consumers have an affective connection with meat (meat attachment) that may play a role in their willingness to change consumption habits [15]. It is argued that the affection towards meat may represent a continuum in which one end refers to disgust (i.e., negative affect and repulsion, related to moral internalization), while the other shows a pattern of attachment (i.e., high positive affect and dependence towards meat, as well as feelings of sadness and deprivation when considering abstaining from meat consumption) that may hinder a change in consumption habits [15]. Likewise, food neophobia, which refers to the reluctance to eat unfamiliar foods [16], may represent a barrier for a transition to a more sustainable diet. According to Apostolidis and McLeay [17], low levels of acceptance of meat substitutes have been associated with high levels of the construct food neophobia.

For increasing the share of plant proteins in the diet, there are several options. An approach could be the usage of textured soy protein, mushrooms, wheat gluten, pulses, etc., as a complete substitute for animal protein. Another opportunity is to replace only a fraction of the meat product (e.g., 20% to 50%) with plant-based proteins [18]. As mentioned in many countries, consumers are highly attached to meat and consider it as an essential and integral element of their daily diet [15]. So-called meathybrids may be an option for the broad consumer segment that is not interested in totally vegan or vegetarian alternatives to meat. Therefore, meathybrids could serve as a low-threshold offer for this group, facilitating the transition in the direction toward a more healthy and sustainable diet. In this context, it has to be mentioned that consumer preferences are in particular affected by the products’ sensory characteristics. An inferior or low sensory quality can constitute a critical barrier for the market entry of meat substitutes [19,20]. Therefore, meat substitutes, respectively meathybrids, must catch up with real meat products concerning sensory characteristics.

As with many novel technologies, consumers’ lack of understanding of hybrid meat products may lead to scepticism and ultimately to the rejection of these. Through early integration of consumer demand and preferences into the development process, more suitable hybrid products can be designed. Understanding the decision-making process will help to develop tailored communication messages that highlight its benefits as a sustainable and healthy alternative to regular meat products.

The study aims at identifying consumer attitudes and preferences for meat alternatives such as meathybrids. Based on a concise literature overview, a representative online survey was carried out in Germany and Belgium including a Discrete Choice Experiment (DCE) for four product categories (meat balls, chicken nuggets, salami, and mortadella).

## 2. Data Collection and Methods of Data Analysis

Consumer data were collected using a quantitative online survey approach. The respondents were panellists and were recruited by the market research company Savanta (London, UK). The questionnaire comprised questions about general meat consumption, on the one hand, and specific questions concerning preferences for meat substitutes, on the other.

So-called choice experiments were integrated in the survey for measuring the importance and preference of different levels of plant-based protein shares in mortadella, salami, chicken nuggets, and meat balls. Choice experiments have been shown to reduce social desirability bias [21], as individuals often display socially desirable preferences in surveys [22].

The online survey was carried out in Germany with 500 and in Belgium with 501 respondents. Participants had to be meat eaters, and thus, vegetarians and vegans were sorted out a priori. Furthermore, the participants had to be 50% responsible for food shopping in the household. Concerning the age, respondents had to be in the range of 18 to 69 years. Data collection took place in the time period from 8 November until 19 November 2019 (see Table 1). Both samples are approximately representative in relation to gender and the region of residence (federal states). For the age, we highlight that the age group from 50–59 years was somewhat under-represented, whereas the age group from 60–69 years was over-represented in both countries.

In the Results Section, we report descriptive results. For scale development, Cronbach’s alpha was applied [23]. Furthermore, confirmatory factor analyses were run to confirm the validity of the scales by using the R-package psych [24]. For measuring food neophobia, the Food Neophobia Scale (FNS) of Pliner and Hobden [16] was selected. The wording of the German version was chosen from a study by [25]. Participants answered on a five-point response scale that was verbally and numerically anchored (1 = totally disagree, 2 = disagree, 3 = neither disagree nor agree, 4 = agree, 5 = totally agree). The five-point scale was used instead of the originally used seven-point scale for a better display of the questionnaire on tablets and smartphones. The items indicated with (r) in Table 6 were inversely re-coded. Considering that the inclusion of invalid items creates the risk of invalid conclusion [26], a principal components analysis (varimax rotation, eigenvalues greater than one) was carried out to explain the variability of the FNS followed by a confirmatory factor analysis [27]. For measuring consumers’ meat attachment, participants answered a five-point Meat Attachment Scale (MEAS) [28] that was verbally and numerically anchored (1 = strongly disagree, 2 = disagree, 3 = neither disagree nor agree, 4 = agree, 5 = strongly agree). The items indicated with (r) in Table 5 were inversely re-coded. In this study, the Health Consciousness Scale (HS) in the style of Visschers et al. [29] was selected for measuring the impact of this psychometric construct on the choice of hybrid products.

Furthermore, a multinomial logistic regression model was applied for measuring the impact of several parameters on the the choice of meat and meathybrid products. Data were collected via a Discrete Choice Experiment (DCE).

### 2.1. Discrete Choice Experiment and Experimental Design

The DCE method is based on micro-economic theory according to which consumers always try to maximize their benefit [30]. In DCEs, consumers must choose from a set of different products offered at determined prices. The products differ regarding the tested product attributes (e.g., share of local feed, price, etc.). According to micro-economic theory, participants will choose the product with the highest benefit. By means of DCEs, consumers’ benefit for each tested product attribute can thus be revealed, as well as the influence of each product attribute on the probability of purchasing/choosing the product. In the DCE of this study, the products varied by six attributes: plant-based protein share, EU organic label, origin label for the protein source, environmental claim, nutritional label, and price (see Table 2). The EU organic label was included since previous studies had shown the importance of this aspect to consumers. The five price levels used in the choice experiment were within the price range that encompassed observed market prices at food retailers in Germany during the winter of 2018/1019. The reported attributes and attribute levels were used for generating the experimental design of the choice experiment. The DCE was carried out for four product categories (meatballs, mortadella, salami, chicken nuggets) on the basis of the same underling experimental design structure.

In each choice set, consumers had a choice between four product alternatives and a no-choice option. The no-choice option was included to get a more realistic purchase situation and thus raise the validity of the data [31]. Furthermore, there was always one 100% meat option and one vegetarian option in the sets, whereas for two options, the plant-based protein share varied between 50% and 20%.

A D-efficient unlabelled design (0.949) was generated using the software Ngene [32], and for each product category, eight choice sets were generated. Thus, in total, there were 32 choice sets. The priors used were based on expert judgement and the literature.

Each participant received two choice sets from each product category and thus had to answer in total eight choice sets. The survey order of the choice sets of the alternatives was randomised to prevent ordering effects [33]. The products, respectively the characteristics, are depicted in photographs (see Table 3).

### 2.2. Multinomial Logistic Regression

Multinomial logistic regression is the regression analysis to conduct when the dependent variable is nominal with more than two levels. It is used to model nominal outcome variables, in which the log odds of the outcomes are modelled as a linear combination of the predictor variables. The multinomial logistic model belongs to the family of generalized linear models and as mentioned is used when the response variable is a categorical variable. Suppose that variable $Y_{i}$ represents the offered alternatives in a choice experiment (e.g., the choice between meat and meathybrid), with *i* = 1, …, *n*, and *n* is the number of possible product alternatives. In the case that *n* equals 2, *Y* has outcomes $Y_{1}$ and $Y_{2}$. Both the counts of $Y_{1}$ and $Y_{2}$ follow a binomial distribution. The probability of occurrence of $Y_{1}$ is $\pi_{1}$ and that of $Y_{2}$ is $\pi_{2}$. Logistic regression relates probability $\pi_{1}$ to a set of predictors using the logit link function:(1)$$logit\left(\pi_{1}\right)=ln\left(\frac{\pi_{1}}{\pi_{2}}\right)=ln\left(\frac{\pi_{1}}{1-\pi_{1}}\right)=x^{\prime }\beta$$
where **x** is a vector of predictors (e.g., FNS, MEAS or buying frequency of organic meat) and β is a vector of model coefficients that are typically estimated by maximum likelihood. Equation (Equation 1) can be rewritten as:(2)$$\left(\frac{\pi_{1}}{1-\pi_{1}}\right)=exp\left(x^{\prime }\beta \right)=exp\left(\eta \right)$$

The quotient in Equation (Equation 2) is referred to as the odds. From Equation (Equation 2), it follows that:(3)$$\pi_{1}=\frac{exp(\eta )}{1+exp(\eta )}$$

The binomial logistic regression model is easily generalized to the multinomial case. If there are *n* product alternatives, there are also *n* variables $Y_{1}$, …, $Y_{n}$ with corresponding probabilities of occurrence $\pi_{1}$, …, $\pi_{n}$. Analogous to binomial logistic regression, the odds $\pi_{1}$/$\pi_{n}$, …, $\pi_{n}-1$/$\pi_{n}$ are modelled by means of $exp(\eta_{1})$, …, $exp(\eta_{n-1})$. From $\sum_{i=1}^{n}\pi_{i}=1$, it follows that:(4)$$\pi_{1}=\frac{exp(\eta_{i})}{exp\left(\eta_{1}\right)+exp\left(\eta_{2}\right)+\ldots +exp\left(\eta_{n}\right)}$$
where $exp(\eta_{n})$ = 0. This model ensures that all probabilities are in the interval [0, 1] and that the probabilities sum to 1.

In this paper, the dependent variable is taken from the DCE where respondents had to indicate if they would buy/choose one out of the four offered options or none of these options. The FNS, HS and MEAS, as well as other parameters (e.g., FAMILIARITY= buying frequency of meat substitutes) entered the regression analysis as independent variables. In addition, all three scales were interacted with the different levels of the attribute “plant-based protein share” for analysing their effect on meat, hybrids and the vegetarian alternative.

Given the theoretical background, an model was built according to the following expression:(5)$$\begin{array}{cc} x^{\prime }\beta =\; & \mathrm{meat}*\beta_{1}+\left(\mathrm{meat}+50\mathrm{plant}\right)*\beta_{2}+\left(\mathrm{meat}+35\mathrm{plant}\right)*\beta_{3}+\left(\mathrm{meat}+20\mathrm{plant}\right)*\beta_{4}\\ \; & +\mathrm{reduced}\;{\mathrm{CO}}_{2}*\beta_{5}+\mathrm{organic}*\beta_{6}+\mathrm{Ger}/\mathrm{Bel}\;\mathrm{origin}*\beta_{7}+\mathrm{local}\;\mathrm{origin}*\beta_{8}\\ \; & +\mathrm{high}\;\mathrm{in}\;\mathrm{fibre}*\beta_{9}+high\;of\;nsf.acids*\beta_{10}+price*\beta_{11}\\ \; & +HS*meat*\beta_{12}+HS*\left(meat+50plant\right)*\beta_{13}+HS*\left(meat+35plant\right)*\beta_{14}\\ \; & +HS*\left(meat+20plant\right)*\beta_{15}+FNS*meat*\beta_{16}+FNS*\left(meat+50plant\right)*\beta_{17}\\ \; & +FNS*\left(meat+35plant\right)*\beta_{18}+FNS*\left(meat+20plant\right)*\beta_{19}+MEAS*meat*\beta_{20}\\ \; & +MEAS*\left(meat+50plant\right)*\beta_{21}+MEAS*\left(meat+35plant\right)*\beta_{22}+MEAS*\left(meat+20plant\right)*\beta_{23}\\ \; & +FAMILIARITY*meat*\beta_{24}+FAMILIARITY*\left(meat+50plant\right)*\beta_{25}\\ \; & +FAMILIARITY*\left(meat+35plant\right)*\beta_{26}+FAMILIARITY*\left(meat+20plant\right)*\beta_{27}+\mathrm{no}-\mathrm{option} \end{array}$$

From the estimation results, odds ratios are calculated. Odds ratios in logistic regression can be interpreted as the effect of one unit of change in X in the predicted odds ratio with the other variables in the model held constant.

In this study, for estimating the specified model, the software R [34] and the package mlogit [35] were used. For the visualisation of the odds ratios from the estimated model, the package sjplot [36] was applied.

## 3. Results

### 3.1. General Buying Behaviour

At the beginning of the questionnaire, the participants had to indicate where they buy most of their meat products. The classical retailer took the first position (48.6%) followed by discount shops (38.6%). Butcheries were in third position (10.2%). All other options were only of minor importance (see Figure 1).

Furthermore, respondents were asked for their buying frequency of organic, respectively free-range, meat. About 22% of the participants indicated buying such products often (18.2%) or always (4.2%) (see Table 4). In 2019, a survey was conducted by Kitchen Stories investigating the purchasing behaviour towards organic food in Germany. In the mentioned study, somewhat higher values were found with 13.2% buying mostly organic products, while for 18.6% of the respondents, organic food made up more than half of the shopping cart.

### 3.2. Scales: Meat Attachment Scale, Neophobia Food Scale and Health Scale

#### 3.2.1. Meat Attachment Scale

In Germany, due to the confirmatory factor analysis, the item “I would feel fine with a meatless diet” was deleted from the scale because in the four-factor solution, this item had a similar loading on different factors and its deletion increased the calculated indices. The reliability analysis for the global MEAS showed in Germany a high internal consistency with a standardized Cronbach α of 0.86. The Comparative Fit Index (CFI = 0.962), the Tucker–Lewis Index (TLI = 0.952) and the Root Mean Squared Error of Approximation (RMSEA = 0.060) showed acceptable values.

In Belgium, likewise due to the confirmatory factor analysis, the item “I would feel fine with a meatless diet” was deleted from the scale because in the four-factor solution, this item had a similar loading on different factors. The reliability analysis for the global MEAS showed a high internal consistency with a standardized Cronbach α of 0.86. The Comparative Fit Index (CFI = 0.959), the Tucker–Lewis Index (TLI = 0.947) and the Root Mean Squared Error of Approximation (RMSEA = 0.067) showed acceptable values.

In comparison to Graça et al. [15], in both countries, we received higher values for the non-reversed items and lower values for the reversed item, which was due to the fact that vegans and vegetarians were not part of this study (see Table 5). On average, respondents agreed to all of the statements. The highest means were received for the statements “I love meals with meat” (3.94) and the reverse-coded item “Meat reminds me of diseases”. The MEAS findings demonstrates that on average, German and Belgium respondents considered meat not as an unhealthy product, but as an essential part of their diet.

#### 3.2.2. Food Neophobia Scale

For Germany, after deleting two items from the original FNS list due to low item correlations in the reliability analysis and one item due to the confirmatory factor analysis, the FNS showed an acceptable internal consistency with a standardized Cronbach α of 0.76 (see Table 6).

The confirmatory factor analysis (two-factor solution) produced acceptable values for the three considered indices (CFI = 0.961, TLI = 0.937 and RMSEA = 0.074). The deleted items were: “I do not trust new (different or innovative) food”, “If I don’t know what a food is, I won’t try it”, and “I am very particular about the food I eat”. For use in the regression analysis, the individual scores, that is the z-standardised mean value across the seven items, were calculated. The higher the FNS is, the higher is the individuals’ food neophobia.

For Belgium, Item No. 9 had to be deleted due to the findings of the confirmatory factor analysis, and the FNS showed an acceptable internal consistency with a standardized Cronbach α of 0.75. The Comparative Fit Index (CFI = 0.949) and the Tucker–Lewis Index (TLI) (0.929), as well as the Root Mean Squared Error of Approximation (RMSEA) 0.073 showed acceptable values for the two-factor solution.

#### 3.2.3. Health Scale

For the applied Health Scale (HS) in Germany (α = 0.81) and Belgium (α = 0.88), acceptable internal consistencies could be measured (see Table 7). Because the HS consisted only of three items, a CFAwith one factor has zero degrees of freedom. In this case, the model is saturated, and there are no degrees of freedom left over to assess model fit. Nonetheless, due to the high Cronbach α values and high factor loadings (<0.6) in the factor analyses, the developed scale was used for the subsequent analysis.

### 3.3. Consumption and Perception of Substitutes

The survey questionnaire comprehended several direct questions about the consumption of meat substitutes. In this context, respondents were asked if they deliberately substitute meat on the days they did not eat meat. In this context, a high proportion of 54.2% of the respondents stated consciously choosing meatless alternatives (see Table 8).

Subsequently, this group had to indicate with which products they concretely substitute meat. For this purpose, they received a list of twelve products, and from that, up to three products could be chosen. The option fish was selected by 48.3% of this segment in Germany and 66.7% in Belgium, followed by cheese (G: 47.6%, B: 29.6%), eggs (G: 41.7%, B: 58.8%), pasta (G: 39.5%, B: 36.7%) and salad (G: 35.4%, B: 16.7%) as the most preferred substitutes (see Table 9). We highlight that the top three on the list were non-vegan alternatives, whereas vegan alternatives like protein-rich lentils, tofu, or seitan were only of minor importance. Furthermore, the findings correspond with the results of De Boer et al. [2], who found a similar ranking (fish, eggs, cheese, etc.) with lentils, nuts, seitan, tempeh and tofu as less often mentioned items (<20%). From a sustainability perspective, the first ranked products do not offer much advantage compared with meat [37].

Additionally, all respondents were asked how often they buy plant-based meat substitutes, such as veggie burgers. Interestingly, only 4.0% (Germany), respectively 4.4% (Belgium), indicated consuming such products frequently, whereas 14.4%, respectively 16.6%, stated doing so at least sometimes (see Table 10). The figures are somewhat lower as those found by De Boer et al. [2] for the Netherlands, where 8% of the respondents reported buying such products frequently. In contrast, similar values as in Germany were observed by Siegrist and Hartmann [25] for Switzerland, where about 23.5% stated consuming substitutes.

In the study, respondents had to indicate if they considered either meathybrids or meat as tastier. Furthermore, they had to decide which of the alternatives was better for the environment, better for animal welfare and healthier. Concerning the parameters environment and animal welfare, the meathybrid was evaluated as much better than the meat option (see Table 11). Contrarily, meat was perceived as tastier in comparison to the meathybrid by 62.4% of the respondents in Germany and 62.7% in Belgium. Concerning the perceived healthiness, the findings differed between the countries. Whereas in Germany, the hybrid was perceived as healthier, the opposite held for Belgium. We highlight that contrary to the reported literature for the perception of meat substitutes, at least in Germany, meathybrids were on average seen as healthier than meat. This outcome is quite surprising against the background that only respondents that consume meat were interviewed.

### 3.4. Multinomial Logit Regression Analysis

In the multinomial regression analysis, it was explored whether the MEAS, the FNS, the HS and all other analysed parameters had an impact on the decision in the DCE (see Section 3.3 for the applied model). In the DCE, the respondents were directly asked if they would choose one out of the four offered product alternatives or none of these products. On the basis of the choice experiments carried out, four logistic regression models for the product categories mortadella, chicken nuggets, salami and meat balls were calculated for Germany and four models for Belgium (see Table 12). In the estimation models, the vegetarian option was set as the reference category for the estimation against the alternatives meat, meat + 50%-plant-based protein seed, meat + 35% plant-based protein seed and meat + 20%-plant-based protein seed. As expected, the price parameter was negative and significant in all models and with one exemption, whereas the organic label predominantly exerted a positive effect on product choice.

The coefficients for the meat options were all significant and revealed the highest positive values in comparison to all other parameters. Thus, the fact that the product was a pure meat product had the highest relevance in the analysed sample of meat eaters. Nonetheless, we highlight that all coefficients for the meathybrids were positive as well, and out of the 24 coefficients for hybrids, fifteen were significant. That is, the vegetarian product was the least preferred in the experiment, whereas the pure meat products had the highest consumer preference, followed by the hybrids. For the plant-based shares of 50%, 35% and 20% in the meathybrids, no real preference order can be stated. Dependent on the country and the product category, sometimes, the plant-based share of 50% was the most preferred option (e.g., nuggets in Germany = 0.930 ***) and, sometimes, the lowest share (e.g., meat balls in Belgium = 0.837 *). Nonetheless, it can be generalised that the hybrids performed better in the DCE compared to the vegetarian alternative. Furthermore, the previous use of meat substitutes had a positive impact on the choice of hybrids in particular for a 50% plant-based protein share (seven out of eight cases). Contrarily, this parameter had a negative impact on the choice of the meat alternatives in the product categories mortadella, salami and meat balls.

Concerning the environmental label “reduced CO_2_”, six out of the eight coefficients were positive and significant. Thus, the use of such a label on the product packaging for hybrids can be recommended. For the applied health labels, this holds only for the product category chicken nuggets and the claim “high of non-saturated acids”, whereas for the other products, no such effect could be measured. Across all products and countries, the local origin had a positive effect on product choice (six out of eight parameters were significant), whereas these held for the national labels only for Germany. As expected, the MEAS exerted in both countries a positive impact on the preference of the meat alternatives in all product categories, whereas for the hybrids, there were only a few significant parameters (three out of 24). Therefore, this psychological construct represents a barrier for the consumption of hybrids because it directly increases the preference for the default option “pure” meat. This finding is in line with Graça et al. [38], who found negative significant associations from meat attachment to meat substitution.

Concerning the HS, it can be stated that the lower the health consciousness was, the lower was the preference for meat. Interestingly, for most of the hybrid variants across product categories and countries (19 out of 24 parameters), there were no significant differences between the hybrid variants and the vegetarian alternatives. Thus, it can be concluded that on average, the health conscious segment saw no serious differences in the health characteristics of these options. That is, the vegetarian and hybrids alternatives were seen both as healthier compared to the meat alternative from this segment. For the impact of the FNS on the preference of hybrids, the results were quite mixed. Only nine out of 24 parameters were significant, and no real order or systemic behaviour can be identified. Therefore, the hypothesis, that food neophobia is a barrier for the choice of hybrids, cannot be affirmed. In this context, we point out that in Belgium, the FNS even reduced the choice probability of the pure meat alternatives (three out four cases significant), whereas no such effect can be found in Germany. In Figure 2 and Figure 3, the odds ratios of the estimations are graphically displayed. The figures clearly show that the sample had a far distance from the highest preference for the pure meat alternatives.

## 4. Discussion and Conclusions

The results show that more than fifty percent of consumers substitute meat at least occasionally. Thus, about half of the respondents revealed an eligible consumption behaviour with respect to sustainability and healthiness, at least sometimes. Furthermore, about a fifth indicated sometimes consuming, respectively frequently, meat alternatives such as veggie burgers. However, most of the consumed meat alternatives had an animal origin (dairy products, fish, eggs), which, like meat production, come along with an environmental burden. In this context, we highlight, that the findings of this study demonstrate that at least a substantial amount of consumers are open-minded to the “meathybrid” concept. Even a higher share believes that this new alternative is better for the environment and the animals in comparison to meat. In the DCE, the tested meathybrid variants with different shares of plant-based proteins took the second position, followed by the vegetarian-based alternative. Therefore, meathybrids could facilitate the diet transition of meat-eaters in the direction toward a more sustainable consumption. The analysed consumer segment was more open-minded to the meathybrid concept in comparison to the vegetarian substitutes. Thus, there is chance that hybrids could serve as a low-threshold option for a transition in the direction toward a more sustainable diet.

Nonetheless, on the road-map of this transition, some major problems and issues have to be tackled. Other research [39,40] suggested that the current meathybrids and meat replacers are still relatively unfamiliar and that their image in relation to the expected taste as shown in this study is quite mixed. While meathybrids have a plant-based protein share, they are not necessarily optimal from an environmental perspective, because their processing stage can require a considerable input of energy, and they often contain a high share of egg protein [41,42]. Although technological development has led to improvements in the quality of meathybrids in recent years, it is still important to develop improved meathybrids and meat substitutes, which are significantly better and superior compared to meat in several ways, such as taste, texture and environmental performance with a lower input of energy and egg whites [43,44].

Despite the technological challenges, there are cultural, respectively socio-cultural challenges as well. As shown, meat attachment as a psychological construct represents a barrier for diet change and transition. Future research should address this topic in more detail and analyse how to overcome this attitude.

Concerning the impact factors on choosing either meathybrids or meat, it becomes obvious that familiarity with meat substitutes, respectively their former use, plays a great role in preference formation. Therefore, it can be recommended to increase the share of meat substitutes/meathybrids in school/public canteens and to financially support other canteens that replace meat with plant substitutes or hybrids. Herewith, consumers are confronted more often with meat alternatives, and familiarity with such products can be built up on a mid- to long-term time horizon.

## Figures and Tables

**Figure 1 foods-10-00071-f001:**
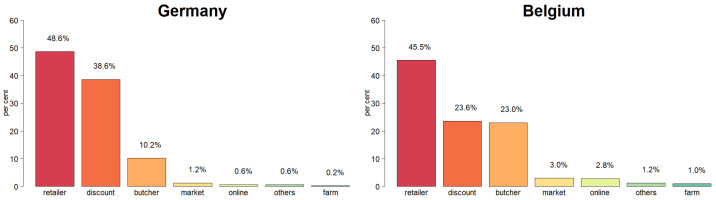
Preferred buying location of meat/meat products.

**Figure 2 foods-10-00071-f002:**
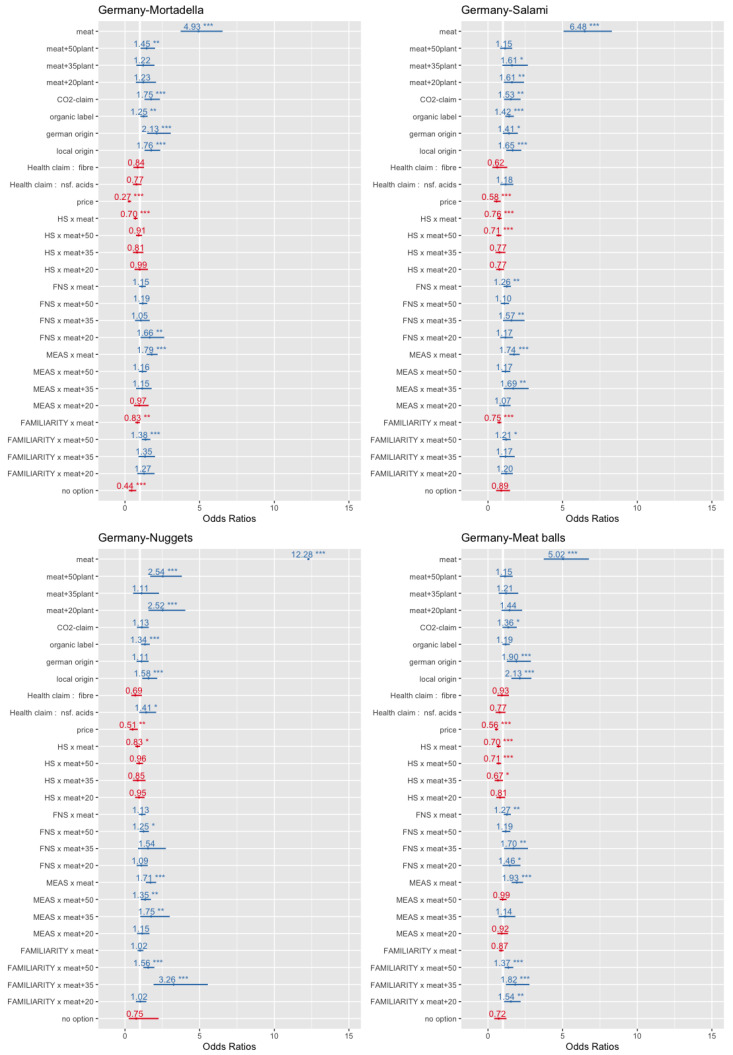
Odds ratios—estimations for Germany. * *p* < 0.1; ** *p* < 0.05; *** *p* < 0.01.

**Figure 3 foods-10-00071-f003:**
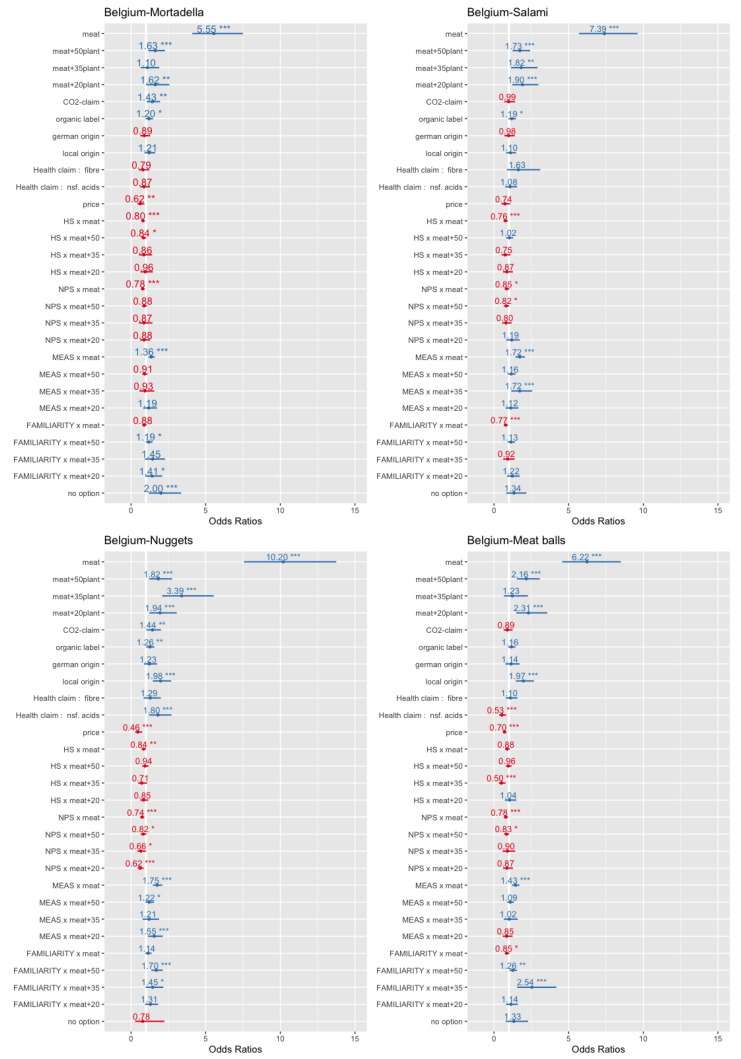
Odds ratios—estimations for Belgium. * *p* < 0.1; ** *p* < 0.05; *** *p* < 0.01.

**Table 1 foods-10-00071-t001:** Sample.

			Germany			Belgium		
			Sample	pop. *		Sample	pop. *
Attribute	Characteristics	*n*	%	%	*n*	%	%
gender	male	245	49.0	49.5	245	48.9	50.0
	female	255	51.0	50.5	256	51.1	50.0
federal state	Baden-Württemberg |Bruxelles	66	13.2	12.9	62	12.4	10.8
	Bayern | Brabant wallon	75	15.0	15.9	27	5.4	3.5
	Berlin | Hainaut	22	4.4	4.4	68	13.6	11.7
	Brandenburg | Liège	15	3.0	3.0	44	8.8	9.7
	Bremen | Luxembourg	5	1.0	0.8	13	2.6	2.5
	Hamburg | Namur	10	2.0	2.2	27	5.4	4.4
	Hessen | Antwerpen	37	7.4	7.4	88	17.6	16.2
	Mecklenburg-Vorp.| Provincie Limb.	10	2.0	1.9	20	4.0	7.8
	Niedersachsen | Oost-Vlaand.	50	10.0	9.7	58	11.6	10.6
	Nordrhein-Westfalen | Vlaams-Brab.	114	22.8	22.0	60	11.9	10.0
	Rheinland-Pfalz | West-Vlaand.	20	4.0	5.1	34	6.8	10.3
	Saarland	5	1.0	1.2		
	Sachsen	26	5.2	4.9		
	Sachsen-Anhalt	15	3.0	2.6		
	Schleswig-Holstein	15	3.0	3.4		
	Thüringen	15	3.0	2.6			
age	18–29 years	95	19.0	20.6	106	21.2	19.3
	30–39 years	88	17.6	18.7	98	19.6	20.3
	40–49 years	84	16.8	18.4	106	21.2	20.6
	50–59 years	88	17.6	23.9	76	15.2	21.8
	60–69 years	145	29.0	18.3	115	23.0	18.1
education	no school qualifications	2	0.4		22	4.4	
	still in school	4	0.8		18	3.6	
	junior high diploma	88	17.6		20	4.0	
	high school diploma	193	38.6		229	45.7	
	university-entrance diploma	105	21.0		78	15.6	
	bachelor’s or master’s degree	89	17.8		122	24.4	
	other degree	19	3.8		12	2.4	
net income	no income	26	5.2		39	7.8	
	less than 500€	30	6.0		19	3.8	
	500€ up to 1000€	46	9.2		36	7.2	
	1000€ up to 1500€	95	19.0		98	19.6	
	1500€ up to 2000€	92	18.4		115	23.0	
	2000€ up to 2500€	69	13.8		89	17.8	
	2500€ up to 3000€	57	11.4		38	7.6	
	3000€ up to 3500€	27	5.4		27	5.4	
	3500€ up to 4000€	25	5.0		23	4.6	
	4000€ or more	33	6.6		17	3.4	
household size	1	121	24.2		112	22.4	
	2	207	41.4		164	32.7	
	3	92	18.4		96	19.2	
	4	55	11.0		82	16.4	
	5	20	4.0		33	6.6	
	6	4	0.8		9	1.8	
	>6	1	0.2		5	1.0	

* Sources: www.statbel.fgov.be and b4p2019 I—Strukturanalyse (www.gik.media/best-4-planning).

**Table 2 foods-10-00071-t002:** Attributes and attribute levels used in the Discrete Choice Experiment (DCE).

Attributes	Levels
plant-based protein share	100% (vegetarian), 50%, 35%, 20%, 20, 0% (meat)
EU organic label	yes, no
origin label prot.source	locally produced produced in Ger/Belg no indicated origin
environmental claim	20% reduced carbon foot print no indicated claim
nutritional label	high content of non-saturated fatty acids high in fibre no indicated label
price	high, middle, low

**Table 3 foods-10-00071-t003:** Choice set example—meat balls.

	Meat Ball 1	Meat Ball 2	Meat Ball 3	Meat Ball 4
	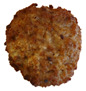	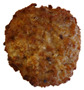	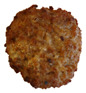	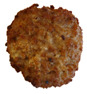
	300 g	300 g	300 g	300 g
Ingredients (plant-based protein share)	100% pork	50% pork + 50% plant-based protein seed	65% pork + 35% plant-based protein seed	100% vegetarian
Organic label	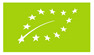			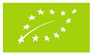
Price	3.29€	2.29€	2.29€	2.29€
Origin of meat, resp. plant-based protein source	Locally produced			Produced in Germany
Environmental claim		20% reduced carbon foot print		

**Table 4 foods-10-00071-t004:** Buying frequency of organic/free-range meat.

	Germany	Belgium
never	24.4%	15.4%
sometimes	57.7%	62.2%
often	15.2%	18.2%
always	3.0%	4.2%

**Table 5 foods-10-00071-t005:** Meat Attachment Scale (MEAS) questionnaire.

	Germany			Belgium		
Statement	std. α	$\bar{x}$	σ	std. α	$\bar{x}$	σ
I love meals with meat.	0.84	3.94	1.00	0.84	3.69	1.03
To eat meat is one of the good pleasures in life.	0.85	3.38	1.08	0.84	3.36	1.10
I’m a big fan of meat.	0.84	3.58	1.07	0.84	3.58	1.07
A good steak is without comparison.	0.84	3.76	1.12	0.85	3.43	1.13
By eating meat I’m reminded of the death and suffering of animals. (r)	0.86	3.50	1.19	0.87	3.42	1.25
To eat meat is disrespectful towards life and the environment. (r)	0.86	3.30	1.19	0.87	3.23	1.12
Meat reminds me of diseases. (r)	0.86	3.86	1.18	0.87	3.70	1.15
To eat meat is an unquestionable right of every person.	0.86	3.57	1.12	0.86	3.60	1.05
According to our position in the food chain, we have the right to eat meat.	0.86	3.68	1.13	0.87	3.70	1.15
Eating meat is a natural and indisputable practice.	0.85	3.75	0.98	0.85	3.58	1.00
I don’t picture myself without eating meat regularly.	0.85	3.56	1.14	0.85	3.44	1.10
If I couldn’t eat meat I would feel weak.	0.85	3.12	1.19	0.85	3.07	1.07
I would feel fine with a meatless diet.		3.32	1.14		3.11	1.11
If I was forced to stop eating meat I would feel sad.	0.85	3.38	1.15	0.85	3.35	1.14
Meat is irreplaceable in my diet.	0.84	3.43	1.11	0.85	3.29	1.02

Note: 5-point Likert scale: 1 = strongly disagree, 2 = disagree, 3 = neither disagree nor agree, 4 = agree, 5 = strongly agree.

**Table 6 foods-10-00071-t006:** Food Neophobia Scale (FNS). (r), inversely re-coded.

	Germany			Belgium		
Statement	std. α	$\bar{x}$	σ	std. α	$\bar{x}$	σ
I am constantly sampling new and different food. (r)	0.74	2.75	1.17	0.74	2.79	1.14
I do not trust new (different or innovative) food.		2.93	1.11	0.73	2.81	1.05
If I don’t know what a food is, I won’t try it.		3.85	1.00	0.74	3.16	1.08
I prefer food from different cultures. (r)	0.72	2.59	1.07	0.75	2.92	1.03
I am reluctant to eat foreign food that I see for the first time.	0.75	2.96	1.21	0.71	2.86	1.17
If I go to a buffet, meetings or parties, I’ll eat new food. (r)	0.73	2.32	1.09	0.73	2.45	0.99
I’m afraid to eat food that I did not eat before.	0.74	2.49	1.23	0.71	2.66	1.18
I am very particular about the food I eat.		2.94	1.13	0.74	3.00	1.26
I will eat almost anything. (r)	0.76	2.32	1.13		2.65	1.20
I like to try new ethnic restaurants. (r)	0.70	2.36	1.10	0.73	2.61	1.07

Note: 5-point Likert scale: 1 = totally disagree, 2 = disagree, 3 = neither disagree nor agree, 4 = agree, 5 = totally agree.

**Table 7 foods-10-00071-t007:** Health Scale (HS).

	Germany			Belgium		
Statement	std. α	$\bar{x}$	σ	std. α	$\bar{x}$	σ
I think it is important to eat healthily	0.74	5.78	1.30	0.83	5.52	1.43
My health is dependent on how and what I eat	0.67	5.38	1.40	0.80	5.29	1.49
If one eats healthily, one gets ill less frequently	0.82	5.33	1.37	0.88	5.14	1.53

Note: 5-point Likert scale: 1 = totally disagree, 2 = disagree, 3 = neither disagree nor agree, 4 = agree, 5 = totally agree.

**Table 8 foods-10-00071-t008:** Deliberate substitution of meat on the days respondents did not eat meat.

	Germany	Belgium
	%	%
yes	54.2	58.7
no	45.8	41.3

**Table 9 foods-10-00071-t009:** Ranking list of consumed meat alternatives.

	Germany		Belgium	
nr	Product	%	Product	%
1	Fish	48.3	Fish	66.7
2	Cheese	47.6	Egg(s)	58.8
3	Egg(s)	41.7	Pasta	36.7
4	Pasta	39.5	Cheese	29.6
5	Salad	35.4	Salad	16.7
6	Other legumes	15.1	Lentils	10.9
7	Lentils	9.6	Nuts	6.5
8	Nuts	8.9	Other legumes	5.4
9	Tofu	6.3	Tofu	5.1
10	Seitan	1.8	Other	2.3
11	Other	1.1	Tempeh	1.0
12	Tempeh	0.4	Seitan	0.7

**Table 10 foods-10-00071-t010:** Frequency of consumption of meat alternatives such as veggie burgers.

	Germany	Belgium
	%	%
never	45.6	41.3
tried it once	16.0	14.6
rarely	20.0	23.2
sometimes	14.4	16.6
frequently	4.0	4.4

**Table 11 foods-10-00071-t011:** Perception meat vs. hybrid.

	Germany			Belgium		
	Meat	Neither/Nor	Hybrid	Meat	Neither/Nor	Hybrid
tastier	62.4%	20.8%	16.8%	62.7%	14.0%	23.4%
healthier	31.0%	27.2%	41.8%	45.3%	14.4%	40.3%
better for environment	15.8%	31.0%	53.2%	22.6%	24.2%	53.3%
better for animal welfare	15.6%	26.8%	57.6%	20.2%	28.9%	50.9%

**Table 12 foods-10-00071-t012:** Estimation results.

	Mortadella		Salami		Nuggets		Meat Balls	
	GER	BEL	GER	BEL	GER	BEL	GER	BEL
	(1)	(2)	(3)	(4)	(5)	(6)	(7)	(8)
meat	1.596 ${}^{***}$	1.713 ${}^{***}$	1.869 ${}^{***}$	2.001 ${}^{***}$	2.508 ${}^{***}$	2.322 ${}^{***}$	1.614 ${}^{***}$	1.828 ${}^{***}$
	(0.143)	(0.154)	(0.126)	(0.134)	(0.158)	(0.152)	(0.152)	(0.158)
meat + 50plant	0.372 ${}^{**}$	0.486 ${}^{***}$	0.137	0.549 ${}^{***}$	0.930 ${}^{***}$	0.601 ${}^{***}$	0.138	0.770 ${}^{***}$
	(0.167)	(0.170)	(0.178)	(0.171)	(0.208)	(0.210)	(0.187)	(0.178)
meat + 35plant	0.202	0.100	0.473 ${}^{*}$	0.598 ${}^{**}$	0.107	1.222 ${}^{***}$	0.187	0.203
	(0.244)	(0.274)	(0.260)	(0.241)	(0.366)	(0.250)	(0.265)	(0.314)
meat + 20plant	0.205	0.482 ${}^{**}$	0.473 ${}^{**}$	0.643 ${}^{***}$	0.925 ${}^{***}$	0.662 ${}^{***}$	0.366	0.837 ${}^{***}$
	(0.269)	(0.236)	(0.208)	(0.227)	(0.241)	(0.233)	(0.236)	(0.222)
reduced CO_2_	0.557 ${}^{***}$	0.355 ${}^{**}$	0.426 ${}^{**}$	−0.014	0.120	0.361 ${}^{**}$	0.311 ${}^{*}$	−0.113
	(0.150)	(0.159)	(0.181)	(0.185)	(0.175)	(0.171)	(0.176)	(0.168)
organic	0.221 ${}^{**}$	0.186 ${}^{*}$	0.348 ${}^{***}$	0.175 ${}^{*}$	0.289 ${}^{***}$	0.234 ${}^{**}$	0.171	0.151
	(0.099)	(0.102)	(0.104)	(0.106)	(0.111)	(0.109)	(0.107)	(0.105)
Ger/Bel origin	0.756 ${}^{***}$	−0.119	0.344 ${}^{*}$	−0.019	0.102	0.205	0.644 ${}^{***}$	0.132
	(0.186)	(0.187)	(0.178)	(0.175)	(0.183)	(0.180)	(0.210)	(0.207)
local origin	0.563 ${}^{***}$	0.190	0.502 ${}^{***}$	0.094	0.457 ${}^{***}$	0.685 ${}^{***}$	0.758 ${}^{***}$	0.678 ${}^{***}$
	(0.150)	(0.150)	(0.154)	(0.153)	(0.160)	(0.156)	(0.158)	(0.155)
High in fibre	−0.170	−0.231	−0.483	0.490	−0.371	0.256	−0.069	0.099
	(0.211)	(0.220)	(0.378)	(0.326)	(0.252)	(0.222)	(0.208)	(0.182)
High of nsf.acids	−0.265	−0.136	0.167	0.073	0.347 ${}^{*}$	0.590 ${}^{***}$	−0.257	−0.633 ${}^{***}$
	(0.188)	(0.193)	(0.185)	(0.186)	(0.199)	(0.207)	(0.212)	(0.216)
price	−1.293 ${}^{***}$	−0.484 ${}^{**}$	−0.544 ${}^{***}$	−0.308	−0.672 ${}^{**}$	−0.786 ${}^{***}$	−0.576 ${}^{***}$	−0.358 ${}^{***}$
	(0.205)	(0.190)	(0.197)	(0.194)	(0.265)	(0.256)	(0.100)	(0.093)
HS * meat	−0.362 ${}^{***}$	−0.227 ${}^{***}$	−0.281 ${}^{***}$	−0.273 ${}^{***}$	−0.181 ${}^{*}$	−0.178 ${}^{**}$	−0.351 ${}^{***}$	−0.132
	(0.099)	(0.082)	(0.100)	(0.093)	(0.098)	(0.089)	(0.098)	(0.089)
HS * meat + 50plant	−0.089	−0.175 ${}^{*}$	−0.343 ${}^{***}$	0.022	−0.045	−0.060	−0.345 ${}^{***}$	−0.039
	(0.113)	(0.099)	(0.121)	(0.117)	(0.121)	(0.117)	(0.114)	(0.104)
HS * meat + 35plant	−0.211	−0.149	−0.266	−0.287	−0.163	−0.336	−0.404 ${}^{*}$	−0.701 ${}^{***}$
	(0.215)	(0.253)	(0.215)	(0.200)	(0.247)	(0.209)	(0.207)	(0.223)
HS * meat + 20plant	−0.011	−0.038	−0.266	−0.144	−0.056	−0.167	−0.214	0.040
	(0.224)	(0.212)	(0.173)	(0.192)	(0.168)	(0.161)	(0.183)	(0.187)
FNS * meat	0.137	−0.243 ${}^{***}$	0.233 ${}^{**}$	−0.164 ${}^{*}$	0.126	−0.301 ${}^{***}$	0.241 ${}^{**}$	−0.245 ${}^{***}$
	(0.099)	(0.080)	(0.101)	(0.093)	(0.101)	(0.092)	(0.095)	(0.089)
FNS * meat + 50plant	0.175	−0.126	0.091	−0.204 ${}^{*}$	0.221 ${}^{*}$	−0.201 ${}^{*}$	0.173	−0.188 ${}^{*}$
	(0.114)	(0.099)	(0.130)	(0.112)	(0.128)	(0.118)	(0.118)	(0.102)
FNS * meat + 35plant	0.051	−0.141	0.450 ${}^{**}$	−0.222	0.433	−0.423 ${}^{*}$	0.531 ${}^{**}$	−0.104
	(0.232)	(0.257)	(0.227)	(0.197)	(0.293)	(0.223)	(0.233)	(0.232)
FNS * meat + 20plant	0.504 ${}^{**}$	−0.128	0.159	0.170	0.085	−0.486 ${}^{***}$	0.375 ${}^{*}$	−0.140
	(0.236)	(0.193)	(0.182)	(0.193)	(0.172)	(0.163)	(0.205)	(0.192)
MEAS * meat	0.580 ${}^{***}$	0.307 ${}^{***}$	0.554 ${}^{***}$	0.544 ${}^{***}$	0.535 ${}^{***}$	0.560 ${}^{***}$	0.658 ${}^{***}$	0.359 ${}^{***}$
	(0.103)	(0.081)	(0.102)	(0.094)	(0.103)	(0.091)	(0.101)	(0.089)
MEAS * meat + 50plant	0.151	−0.097	0.156	0.144	0.303 ${}^{**}$	0.200 ${}^{*}$	−0.013	0.083
	(0.115)	(0.098)	(0.126)	(0.113)	(0.126)	(0.117)	(0.118)	(0.101)
MEAS * meat + 35plant	0.142	−0.068	0.523 ${}^{**}$	0.543 ${}^{***}$	0.560 ${}^{**}$	0.190	0.132	0.023
	(0.224)	(0.259)	(0.245)	(0.203)	(0.274)	(0.225)	(0.242)	(0.226)
MEAS * meat + 20plant	−0.032	0.170	0.067	0.114	0.139	0.439 ${}^{***}$	−0.087	−0.163
	(0.251)	(0.196)	(0.178)	(0.192)	(0.181)	(0.162)	(0.195)	(0.192)
FAMILIARITY * meat	−0.188 ${}^{**}$	−0.125	−0.290 ${}^{***}$	−0.256 ${}^{***}$	0.016	0.135	−0.135	−0.157 ${}^{*}$
	(0.096)	(0.082)	(0.094)	(0.091)	(0.100)	(0.093)	(0.092)	(0.092)
FAMILIARITY * meat + 50pl.	0.321 ${}^{***}$	0.173 ${}^{*}$	0.195 ${}^{*}$	0.124	0.442 ${}^{***}$	0.529 ${}^{***}$	0.316 ${}^{***}$	0.233 ${}^{**}$
	(0.106)	(0.096)	(0.116)	(0.108)	(0.120)	(0.116)	(0.108)	(0.102)
FAMILIARITY * meat + 35pl.	0.299	0.369	0.160	−0.086	1.181 ${}^{***}$	0.371 ${}^{*}$	0.601 ${}^{***}$	0.931 ${}^{***}$
	(0.204)	(0.231)	(0.216)	(0.209)	(0.272)	(0.205)	(0.215)	(0.254)
FAMILIARITY * meat + 20pl.	0.242	0.343 ${}^{*}$	0.185	0.201	0.019	0.267	0.429 ${}^{**}$	0.128
	(0.224)	(0.199)	(0.165)	(0.174)	(0.173)	(0.167)	(0.180)	(0.175)
no option	−0.826 ${}^{***}$	0.692 ${}^{***}$	−0.113	0.290	−0.292	−0.243	−0.330	0.285
	(0.281)	(0.265)	(0.257)	(0.248)	(0.563)	(0.535)	(0.277)	(0.277)
Observations	1000	1002	1000	1002	1000	1002	1000	1002
Log Likelihood	−1220.917	−1366.974	−1201.299	−1272.771	−1177.140	−1261.003	−1227.343	−1310.656

* *p* < 0.1; ** *p* < 0.05; *** *p* < 0.01.

## Data Availability

The data presented in this study are available on request from the corresponding author. The data will be made publicly to a later stage.

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
