# Peer review of "Discrete Choice Analysis of Consumer Preferences for Meathybrids—Findings from Germany and Belgium"

_foods, 2020, doi:10.3390/foods10010071_

Round 1
Reviewer 1 Report
The manuscript deals with a timely and interesting topic using a not-so innovative methodology but with strong results supported by a meticulous programming and background knowledge.
Overall, I've found the paper easy and pleasant to read and follow, despite some minor concerns I would like to highlighted using bullet points:
- line 147: authors used NPS as acronym for the Food Neophobia Scale and this is repeated in other parts of the manuscript besides line 264 where FNS is used. Please amend using only FNS (Food Neophobia Scale).
- It could be of interest to add in table 3 how well the sample represent the population of both Countries.
- 4.2 section: please add a reference for the indexes used to assess the GOF of the CFAs.
- line 287: why no CFA for the Health Scale? Since you've done a great job conducting a CFA for every other scale, I was wondering what happened with this one.
- Discussion section should be improved with a connection of your results with the literature, this seems to be a Conclusion section instead of a discussion.
Author Response
Dear Reviewer, we have been very grateful for the comments and critical remarks.
We rewrote in particular the discussion and conclusion as well as the first two
sections (see comments second reviewer) and tried to address all your concerns
you have had. Furthermore, now we improved the ordering and display of the
tables and figures.
Best regards and thanks a lot for your work,
The corresponding author

Reviewer 2 Report
The needs for meat alternatives has been risen with various environmental reasons. With these reason, this study could be a critical study to understand how to feel to general consumer.
There were a few comments to your study.
Chapter 2 can be included in introduction more briefly. The previous studies do not showed in separated chapter.
You should re-ordered and move Table and Figure according to manuscript. Table numbers are out of order.
Manuscript is not double spacing.
Discussion:
Is occasional consumption of substitute really eligible consumption behavior in respect to healthiness? The essential amino acids which is not contained in vegetable sources are abundant in animal sources. Therefore, essential amino acid index could explain the healthiness of meat alternatives.
Furthermore, your discussion seems to repetition of results description. Therefore, you should re-write the discussion with comparison with previous studies and improve the novelty of your study.
Thank you.
Author Response
Dear Reviewer, we have been very grateful for the comments and critical remarks.
We rewrote in particular the discussion and conclusions as well as the first two
sections and tried to address all the concerns you have had. Furthermore, now
we improved the ordering and display of the tables and figures.
Best regards and thanks a lot for your work,
The corresponding author
-----------------------------------------------------------------------------------------------

Round 2
Reviewer 1 Report
I would like to thank the authors for their work, all my concerns have been followed.
Happy holidays!
Reviewer 2 Report
This manuscript was greatly improved and no more comments.